# Uncovering the Genetic and Molecular Features of Huntington’s Disease in Northern Colombia

**DOI:** 10.3390/ijms242216154

**Published:** 2023-11-10

**Authors:** Mostapha Ahmad, Margarita R. Ríos-Anillo, Johan E. Acosta-López, Martha L. Cervantes-Henríquez, Martha Martínez-Banfi, Wilmar Pineda-Alhucema, Pedro Puentes-Rozo, Cristian Sánchez-Barros, Andrés Pinzón, Hardip R. Patel, Jorge I. Vélez, José Luis Villarreal-Camacho, David A. Pineda, Mauricio Arcos-Burgos, Manuel Sánchez-Rojas

**Affiliations:** 1Facultad de Ciencias de la Salud, Universidad Simón Bolívar, Barranquilla 080002, Colombia; mostapha.ahmad@unisimon.edu.co (M.A.); margarita.rios@unisimon.edu.co (M.R.R.-A.); pedro.puentes@unisimon.edu.co (P.P.-R.); doctorcristiansanchez@gmail.com (C.S.-B.); sanchezr@unisimonbolivar.edu.co (M.S.-R.); 2Life Science Research Center, Universidad Simón Bolívar, Barranquilla 080002, Colombia; martha.cervantes@unisimon.edu.co (M.L.C.-H.); martha.martinez@unisimon.edu.co (M.M.-B.); wilmar.pineda@unisimon.edu.co (W.P.-A.); 3Médica Residente de Neurología, Universidad Simón Bolívar, Barranquilla 080002, Colombia; 4Facultad de Ciencias Jurídicas y Sociales, Universidad Simón Bolívar, Barranquilla 080002, Colombia; 5Grupo de Neurociencias del Caribe, Universidad del Atlántico, Barranquilla 080001, Colombia; 6Departamento de Neurofisiología Clínica Palma de Mallorca, Hospital Juaneda Miramar, Islas Baleares, 07011 Palma, Spain; 7Bioinformatics and Systems Biology Laboratory, Institute for Genetics, Universidad Nacional de Colombia, Bogota 111321, Colombia; ampinzonv@unal.edu.co; 8National Centre for Indigenous Genomics, John Curtin School of Medical Research, Australian National University, Canberra, ACT 2601, Australia; hardip.patel@anu.edu.au; 9Department of Industrial Engineering, Universidad del Norte, Barranquilla 081007, Colombia; 10Programa de Medicina, Facultad de Ciencias de la Salud, Universidad Libre Seccional Barranquilla, Barranquilla 081007, Colombia; josel.villarrealc@unilibre.edu.co; 11Grupo de Investigación en Neuropsicología y Conducta, Universidad de San Buenaventura, Medellin 050010, Colombia; david.pineda1@udea.edu.co; 12Grupo de Neurociencias de Antioquia, Universidad de Antioquia, Medellin 050010, Colombia; 13Grupo de Investigación en Psiquiatría (GIPSI), Departamento de Psiquiatría, Instituto de Investigaciones Médicas, Facultad de Medicina, Universidad de Antioquia, Medellin 050010, Colombia; mauricioarcosburgos@gmail.com

**Keywords:** Huntington’s disease, *HTT*, CAG repeats, mosaicism, slippage

## Abstract

Huntington’s disease (HD) is a genetic disorder caused by a CAG trinucleotide expansion in the huntingtin (*HTT*) gene. Juan de Acosta, Atlántico, a city located on the Caribbean coast of Colombia, is home to the world’s second-largest HD pedigree. Here, we include 291 descendants of this pedigree with at least one family member with HD. Blood samples were collected, and genomic DNA was extracted. We quantified the *HTT* CAG expansion using an amplicon sequencing protocol. The genetic heterogeneity was measured as the ratio of the mosaicism allele’s read peak and the slippage ratio of the allele’s read peak from our sequence data. The statistical and bioinformatic analyses were performed with a significance threshold of *p* < 0.05. We found that the average *HTT* CAG repeat length in all participants was 21.91 (SD = 8.92). Of the 291 participants, 33 (11.3%, 18 females) had a positive molecular diagnosis for HD. Most affected individuals were adults, and the most common primary and secondary alleles were 17/7 (CAG/CCG) and 17/10 (CAG/CCG), respectively. The mosaicism increased with age in the participants with HD, while the slippage analyses revealed differences by the HD allele type only for the secondary allele. The slippage tended to increase with the *HTT* CAG repeat length in the participants with HD, but the increase was not statistically significant. This study analyzed the genetic and molecular features of 291 participants, including 33 with HD. We found that the mosaicism increased with age in the participants with HD, particularly for the secondary allele. The most common haplotype was 17/7_17/10. The slippage for the secondary allele varied by the HD allele type, but there was no significant difference in the slippage by sex. Our findings offer valuable insights into HD and could have implications for future research and clinical management.

## 1. Introduction

Huntington’s disease (HD) is an autosomal dominant neurodegenerative disorder first described in 1872 [1]. It is caused by a mutation in the IT15 (interesting transcript 15) (Huntingtin, *HTT*) gene, which harbors an expanded CAG trinucleotide, encoding a polyglutamine domain (polyQ) in its first exon [2]. According to the number of CAG repeats, four clusters of allele-associated phenotypes have been identified [3]: the normal (with alleles carrying ≤26 repeats), the intermediate (27–35 repeats) [4], and the HD-causing pathogenic alleles (≥36 repeats). The HD-causing alleles are further subdivided into the HD-causing alleles with reduced penetrance (36–39 repeats) and the HD-causing alleles with full penetrance (≥40 repeats) [5,6]. The CAG expansion is unstable, and the size of the repeat sequence varies in the germline and somatic cell lineages [3]. Evidence suggests that both sexes are equally affected, and the risk of transmitting the gene on to the next generation is 50%. Furthermore, all individuals who inherit the mutated allele and live long enough will eventually exhibit signs of the disease [4].

More than a quarter of a million Americans suffer with HD [1]. Interestingly, the world’s most extensive pedigree segregating HD inhabits the state of Zulia, Venezuela [5], followed in size by another pedigree inhabiting Juan de Acosta, a city located in the department of Atlántico on the Caribbean coast of Colombia [6]. The first neuroepidemiological study conducted in Colombia in 1991, using an extensive clinical evaluation found in Juan de Acosta, found a high prevalence of some neurological affections, i.e., the presence of 43 cases with probable HD and other common conditions such epilepsy and vascular diseases [7]. In 1991, Juan de Acosta’s HD prevalence was 3.8 × 1000 inhabitants. Nowadays, the estimated prevalence of HD in the department of Atlántico is 0.2 × 10,000 inhabitants, while in Juan de Acosta it is 9.7 × 10,000 inhabitants [7].

Historical records suggest that the founder effect of this HD cluster originated around 1832, when the Spaniard Lucas Echeverría arrived from Cartagena and settled in Juan de Acosta, where he married Josefa Arteta. Four children were born of this union, and according to Josefa, Mr. Echeverría died, weakened by the effects of HD [6].

In this article, we present a molecular and genetic epidemiological assessment of the community of Juan de Acosta with a complete ascertainment of asymptomatic individuals traced to pedigrees segregating HD. Our overall goal is the early detection of HD in an attempt to positively alter the natural history of HD and provide the best genetic counseling to these families [8]. Furthermore, we aimed to evaluate population and epidemiological metrics so that the official bodies in charge of public health can make the best executive decisions. This comprehensive epidemiological evaluation will favor a better clinical management of patients and may therefore lead to a significant improvement in the quality of life of families with HD.

## 2. Results

### 2.1. Demographic Findings

Our sample consisted of 291 participants (167 [57.4%] female and 124 [42.6%] males). Of these, 71 (24.5%) were children, 81 (27.9%) were adolescents and young adults (AYA), 122 (42.1%) were adults, and 16 (5.5%) were elderly. The full sociodemographic characteristics of our participants are presented in Table 1. Appendix A of the Appendix A shows the distribution of age by *HTT* allele type.

### 2.2. Molecular Diagnosis and Genotype Distribution

Based on the molecular diagnosis, the number of *HTT* CAG repeats in our sample ranged from 12 to 51, with an average of 21.91 (SD = 8.92) copies. Of the 291 participants, 16 (5.5%) carried intermediate HD alleles, one (0.3%) carried reduced penetrance HD alleles, and 33 (11.3%) carried full penetrance HD alleles (Table 1). Thus, according to the number of *HTT* CAG repeats, 33 individuals (18 females and 15 males) received a positive molecular diagnosis for HD. Of these individuals, eight (24.2%) were children, four (12.1%) adolescents, 19 (57.6%) adults and two (6.1%) elderly adults. Interestingly, 95% of the affected individuals had at least 44 HTT CAG repeats (Appendix A). In addition, two clusters of individuals were identified when analyzing the distribution of age by the number of *HTT* CAG repeats (Appendix A).

ScaleHD reports the primary and secondary alleles as *a*/*b*, where *a* is the number of CAG repeats and *b* is the number of CCG repeats in the *HTT* gene. Thus far, in ScaleHD, the objects are sorted by a total number of aligned reads—the top allele is always taken as a “primary” allele (where the word “primary” does not have any biological meaning; it is simply the allele primarily chose from the assembly) [9]. The secondary allele, by design, refers to the second allele. It is important to clarify that we refer to them as “genotype” only because of the ScaleHD output “Primary/Secondary GTYPE” [9]. Still, it is not the genotype since the genotype refers to both alleles.

The analysis of the genotype distribution of the primary and secondary alleles shows that 17/7 is the most common primary allele (i.e., 17/7 means 17 CAG repeats with 7 CCG repeats), followed by 15/7, 17/10, and 18/7 (Figure 1a). As for the secondary allele, the most common genotype is 17/10, followed by 17/7, 18/9, and 23/7 (Figure 1a). When examining the allele combinations (genotypes; Appendix A), we identified that the most common haplotype is 17/7_17/10 (*n* = 19, 12.3%), followed by the haplotypes 17/7_17/7 (*n* = 17, 11%) and 15/7_17/7 (*n* = 13, 8.4%) (Figure 1b). Following the ScaleHD protocol, we identified that 268 (92.1%) individuals have a typical sequence structure in the primary allele, 231 (79.4%) individuals have a typical sequence structure in the secondary allele, and 226 (77.6%) individuals have a typical sequence structure in both alleles (Appendix A).

### 2.3. Heterogeneity Analysis

Following the results provided by ScaleHD, we assessed the genetic heterogeneity using two seemingly unrelated approaches.

The first approach, reported in ScaleHD as “Somatic Mosaicism” (i.e., a term that does not mean the real mosaicism, as the DNA analyzed in this study is derived exclusively from hematopoietic cells), corresponds to the ratio of the mosaicism allele’s read peak, and is calculated as *N* + 1 to *N* + 10 over *N*, where *N* is the number of reads; these quantities represent an addition of repeats over the total number of reads. We found that, among the HD-affected individuals, the mosaicism ratio tends to increase with age, and this pattern is statistically significant only for the second allele (*r* = 0.349, *t*_31_ = 2.075, *p* = 0.046; Figure 2a). However, no correlation between mosaicism and age was observed in the HD-unaffected individuals (primary allele: *r* = 0.071, *t*_255_ = 1.196, *p* = 0.233; secondary allele: *r* = 0.056, *t*_255_ = 0.910, *p* = 0.364; Figure 2b). When testing whether the HD allele type influenced the mosaicism for the primary and secondary genotypes, we found that the mosaicism in the primary (χ^2^ = 42.435, *df* = 3, *p* < 0.00001) and secondary (χ^2^ = 380. 41, *df* = 3, *p* < 0.00001) alleles differed by the HD allele type (Figure 2c), but not by sex (primary allele: χ^2^ = 0.0381, *df* = 1, *p* = 0.845; secondary allele: χ^2^ = 0.0176, *df* = 1, *p* = 0.895; Figure 2d).

In the second approach, we used the BSlippage reported by ScaleHD, which corresponds to the slippage ratio of the allele’s read peak and is calculated as *N* − 2 to *N* − 1 over *N*, where *N* is the number of repeats; these quantities represent a subtraction of repeats over the total number of reads. In our analyses, we identified that the slippage ratio differed by the HD allele type for the secondary allele (χ^2^ = 24.711, *df* = 3, *p* < 0.0001), but not for the primary allele (χ^2^ = 6.196, *df* = 3, *p* = 0.102) (Figure 3a). No statistically significant difference in the slippage ratio was found by sex (primary allele: χ^2^ = 0.141, *df* = 1, *p* = 0.707; secondary allele: χ^2^ = 2.98, *df* = 1, *p* = 0.084; Figure 3b). On the other hand, although the slippage ratio tends to decrease with age, this pattern is not statistically significant in HD-affected individuals (primary allele: *r* = −0.187, *t*_31_ = −1.062, *p* = 0.296; secondary allele: *r* = −0.106, *t*_31_ = −0.594, *p* = 0.557; Figure 3c). However, this pattern is statistically significant in HD-unaffected individuals for the secondary allele (*r* = 0.151, *t*_255_ = 2.367, *p* = 0.015), but not for the primary allele (*r* = −0.0616, *t*_255_ = −0.986, *p* = 0.325) (Figure 3c). Finally, we found that the slippage ratio tends to increase with the number of CAG repeats in HD-affected individuals, but this correlation is not statistically significant (primary allele: *r* = 0.067, *t*_31_ = −0.375, *p* = 0.709; secondary allele: *r* = 0.072, *t*_31_ = 0.405, *p* = 0.688; Figure 3d).

Complementary analyses of these heterogeneity measures combining the ‘Somatic Mosaicism’ and BSlippage for both alleles are presented in Figure 4. The MANOVA analyses revealed that the “Somatic Mosaicism” (Figure 4, left) and slippage ratio (Figure 4, right) in both alleles depend on HD diagnosis (‘Somatic Mosaicism’: *F*_2,288_ = 210.12, *p* < 0.001; Slippage Ratio: *F*_2,288_ = 8.84, *p* < 0.001). While the effect on the “Somatic Mosaicism” is present in both alleles (primary allele: *F*_1,289_ = 14.51, *p* < 0.001; secondary allele: *F*_1,289_ = 401.83, *p* < 0.001), in the slippage ratio, it is only present for the secondary allele (*F*_1,289_ = 12.54, *p* < 0.001). Furthermore, we found that the correlation between the slippage ratio in the primary and secondary allele is statistically significant regardless of HD diagnosis (unaffected: *r* = 0.443, *t*_31_ = 7.91, *p* < 0.0001; affected: *r* = 0.913, *t*_31_ = 12.5, *p* < 0.0001), but for “Somatic Mosaicism”, it is not (unaffected: *r* = 0.046, *t*_256_ = 0.739, *p* = 0.461; affected: *r* = 0.163, *t*_31_ = 0.904, *p =* 0.373).

## 3. Discussion

Huntington’s disease (HD) is a progressive, autosomal dominant, neurodegenerative disease that affects the brain, and it is caused by a genetic mutation in exon 1 of the *huntingtin* (*HTT*) gene, located at chromosome 4p16.3. The normal range of CAG repeats in the *HTT* gene is 6 to 26, while people with HD typically have 36 or more CAG repeats. The number of CAG repeats can affect the age of onset and the rate of progression of the disease, with more repeats leading to an earlier onset and more severe symptoms.

In our sample of 291 individuals, the number of *HTT* CAG repeats ranges from 12 to 51, with an average of 21.91 copies (Table 1). Following the international recommendations, we derived four subgroups according to the number of *HTT* CAG repeats: Normal (≤26 copies; *n* = 241, 82.8%), intermediate (27–35 copies; *n* = 16, 16%), reduced penetrance (36–39 copies; *n* = 1, 0.3%), and full penetrance (≥40 copies; *n* = 33, 11.3%) (Table 1).

Previous studies have shown a significant correlation between the average CAG repeat length of normal chromosomes and the prevalence of HD. The average wild-type CAG repeat size was significantly larger in populations with a higher prevalence of HD [10,11]. Therefore, the *HTT* CAG repeat size in a large sample of Colombian subjects may in turn reflect the prevalence of HD in the Colombian population. Table 2 shows the comparison of the number of CAG repeat sizes of normal and intermediate alleles between the current study and other populations. This comparison shows that the average *HTT* CAG repeat sizes between the Colombian and other populations is similar, with the European population being the closest.

The genotype distribution shows that the most common primary genotype is 17/7, followed by 15/7, 17/10, and 18/7; in the secondary genotype, the most common genotype is 17/10, followed by 17/7, 18/9, and 23/7; the genotype combinations (haplotypes) show the most common is 17/7_17/10 (*n* = 19, 12.3%), followed by 17/7_17/7 (*n* = 17, 11%), and 15/7_17/7 (*n* = 13, 8.4%) (Figure 1a). Our results are consistent with previous reports that the average CAG tract size in the East Asian general population was 16.9 repeats and 17.8 repeats in Europeans [11]. This study also shows a correlation with the *HTT* haplogroups of the general population (<27 CAG repeats). The A1 and A2 haplotypes are two of the most common haplotypes associated with the HD mutation. These haplotypes are defined by variations found at three specific markers on the huntingtin gene. A person with the A1 haplotype has a specific set of variations at these three markers, while a person with the A2 haplotype has a different set of variations. There is a diversity of haplogroups found in the general European population, although the CAG expansion is most likely to occur on haplogroup A in this population. Note that haplogroup A, and the variants with the highest risk of CAG expansion in the European population (A1 and A2), are absent in the general populations of China and Japan [10,11,14].

Juan de Acosta is a corregimiento in the Atlántico department on Colombia’s northern Caribbean coast with a Basque founding origin. Historically, several corregimientos in the Atlántico department have different ancestral origins, with an ethnic composition based on migratory flows over the years. This event would confirm that the founding mutation in this area occurred in Western Europe and spread to other regions through migration. Furthermore, the CCG7 allele is the predominant allele in Western Europe and could generate variations in the number of CAG repeats through independent mutational events. This finding is consistent with the population studied [15]. In 2020, a study of the CAG intermediate *HTT* alleles in the general population of Rio de Janeiro, Brazil, compared with a sample of families affected by HD, showed that CCG7 was the most frequent allele [16]. On the other hand, the haplotypic analysis of CAG and CCG was repeated in 21 Brazilian families with HD. In total, 40 different haplotypes were identified. Further analysis showed that CCG10 was linked to a normal CAG allele in 19 haplotypes and to expanded alleles in two haplotypes. In addition, CCG7 was linked to expanded CAG repeats in 40 haplotypes (95.24%) and CCG10 was linked to expanded CAG repeats in only two haplotypes (4.76%). Therefore, the CCG7 allele was the most common allele on HD chromosomes in this Brazilian sample, which is consistent with the results obtained in another Brazilian sample [17] and is also consistent with the results obtained in our Caribbean sample.

In 2015, researchers analyzed the CCG repeat polymorphism located near the CAG repeat and identified HD chromosome haplotypes. Surprisingly, the results revealed a strong linkage disequilibrium between the CAG repeat expansion and the CCG10 allele on Japanese HD chromosomes, which differs from what has been reported in Western populations in the past [18]. These repeats suggest that HD mutations in Asian populations may originate from different ancestral lineages and therefore are associated with a high (CCG7 and CCG10) or low (CCG6 and CCG11) prevalence of HD. For example, in the Caucasian population, the CCG11 allele is less prevalent in individuals with HD. Conversely, populations of Western European descent, which have a higher prevalence of HD, have a higher frequency of the CCG7 allele. In contrast, in populations such as Black African, Japanese, Chinese, and Finnish, in which HD is less common, the most common CCG alleles are CCG11 and CCG6 [19]. On the other hand, the frequency and distribution of the HD mutation in Caribbean populations may vary depending on factors such as ancestry, migration patterns, and population history. Indeed, some Caribbean populations, such as those in Jamaica and Trinidad and Tobago, have been reported to have a higher frequency of HD than other populations of African descent [14,20].

The presence of somatic mosaicism in HD can pose challenges for genetic testing and counselling because standard testing methods may miss the mutation if it is present in a small proportion of cells. This can lead to false negative results and an inability to accurately estimate the risk of developing HD or passing it on to offspring [21]. In some cases, mosaicism in HD may result in a less severe form of the disease or a delayed onset of symptoms because the number of cells carrying the mutation may be lower [1,22,23]. In other cases, however, the severity and onset of symptoms may be more unpredictable, as the proportion of cells carrying the mutation can vary widely between individuals. Therefore, if mosaicism in HD is suspected, more sensitive testing methods such as repeat primed PCR or Southern blot analysis may be required to detect the mutation. On the other hand, genetic counselling should also consider the potential impact of mosaicism on the disease onset and progression [22,24]. Here, we found that mosaicism tends to increase with age in HD-affected individuals, but not in HD-unaffected individuals (Figure 2b). Furthermore, mosaicism in the primary and secondary alleles is associated with the HD allele type and gender (Figure 2c,d). It is noteworthy that, in reviewing similar research on HD, we did not find any previous studies reporting information on mosaicism and slippage in a pre-symptomatic population at risk of developing HD. In addition, mosaicism in both the primary and secondary alleles depends on the HD diagnosis (Figure 4, left). Future studies may benefit from considering our findings for early diagnosis, follow-up, and the development of potential treatments for HD [21,25].

The CAG repeat is the genetic mutation responsible for HD, and the number of CAG repeats in the *HTT* gene is used to determine a person’s risk of developing the disease. However, PCR amplification can sometimes cause small errors or “slippage” in the number of CAG repeats counted, leading to inaccurate results. Slippage can also lead to false negative or false positive results in HD genetic testing, particularly in cases where the CAG repeat length is close to the diagnostic threshold for the disease. In this study, we found that the secondary allele slippage ratio is associated with the HD allele type and does not differ by gender (Figure 3b). However, the slippage ratio tends to decrease with age regardless of HD diagnosis in our sample (Figure 3c). We also found that slippage tends to increase with the number of CAG repeats in HD-affected individuals (Figure 3d). In addition, the analyses of the multiple dependency of slippage (Figure 4, right) in the primary and secondary allele by HD diagnosis showed that HD diagnosis plays an important role in defining such a dependency. We also identified statistically significant correlations between slippage in the primary and secondary alleles regardless of HD diagnosis. Although slippage has very specific biological meanings [26], the calculation of slippage in ScaleHD involves the subtraction of repeats from the total number of reads, *N*. Thus, more molecular analyses are needed to prove this in our case.

This study could have important clinical implications by promoting interdisciplinary follow-up for people with HD. This approach would incorporate standardized tools, such as motor, neuropsychiatric, and neuropsychological assessments, like those used in the international Enroll-HD study [27]. Combined with biomarkers in blood and cerebrospinal fluid, such as neurofilament studies, these assessments could track HD progression effectively. Additionally, structural resonance neuroimaging with segmentation, as well as the cross-sectional and longitudinal analyses of static and dynamic brain connectivity [28], should be integrated into follow-up protocols. Finally, cognitive neuroscience studies using controlled tasks and brain signal analysis should also be conducted. This comprehensive interdisciplinary approach, coupled with advanced and intelligent diagnostic tools, promises a more thorough and precise evaluation of HD progression [29,30,31,32], ultimately advancing our understanding and treatment of this neurodegenerative condition.

## 4. Materials and Methods

### 4.1. Study Subjects

The study population consisted of 291 individuals, who are descendants of families residing in Juan de Acosta with at least one member affected by HD. Family genealogy was reconstructed through interviews with family members. Participation was voluntary, and the descendants met the inclusion criteria defined for the research: (i) accept and sign an informed consent form, and (ii) belong to a family with at least one member with HD. Individuals with movement disorders other than HD and/or a history of psychiatric disorders were excluded.

### 4.2. Blood Samples and Genomic DNA Extraction

Peripheral whole blood samples (5 mL) were obtained from individuals who agreed to participate in the research and were placed in Vacutainer^®^ EDTA tubes. The samples were stored at 4 °C until analysis. Genomic DNA extraction was performed using DNeasy Blood & Tissue commercial kit (QUIAGEN, Inc., Germantown, MD, USA), which provides a high-purity extraction product for genotyping. The extracted DNA was resuspended in ultrapure water and stored at −20 °C until analysis. DNA concentration and purity were quantified using Qubit™ 2.0 Fluorometer dsDNA HS Assay Kits (ThermoFisher Scientific, Inc., Waltham, MA, USA).

### 4.3. Quantification of HTT CAG repeats

Genomic DNA was sent to iLab at the University of Arizona, USA, where an amplicon sequencing protocol was used. This extensively validated protocol allows sequencing of hundreds of samples in a single MiSeq run. Library preparation and MiSeq sequencing for genotyping were conducted according to the protocol. The sequence encoding the HTT polyglutamine and polyproline tracts was amplified from genomic DNA using MiSeq-compatible PCR primers. The resulting PCR product can be directly sequenced. After PCR, a fraction of each PCR product is pooled and purified using AMPure XP beads. This PCR cleanup step also allows for the removal of primer dimers. The sequencing library is then quality-controlled using Qubit, Bioanalyzer, and qPCR, and sequenced on the MiSeq platform. The MiSeq-compatible PCR primers were designed based on the TruSeq combinatorial dual design with the addition of spacers between the sequencing primer binding site and the locus-specific primer [33,34,35].

### 4.4. Bioinformatic and Statistical Analyses

#### 4.4.1. Quantification of HTT CAG Repeats

It is well-known that the *HTT* CAG repeat is susceptible to various biological phenomena that can lead to genotyping inaccuracies, making precision a difficult task. To quantify *HTT* CAG repeats in our sample, we used ScaleHD version 1.0 [9]. ScaleHD is an automated HD genotyping bioinformatics pipeline used in large-scale automated genotyping of parallel sequencing data of the *HTT* CAG/CCG repeats associated with HD. Unlike conventional software offering a generalized approach to profiling disease-associated repeat loci, ScaleHD offers an automated, unsupervised solution that ensures more accurate and reliable genotyping results. As part of the automated flow, ScaleHD performs sequence quality control, sequence alignment, and automated genotyping on all FASTQ file pairs provided by the user as input. Once a stage has completed, required information is automatically passed to the next stage [9]. The full set of parameters used to run ScaleHD on our sequence data is reported in Appendix A of the Appendix A.

#### 4.4.2. Statistical Analyses

Demographic and genetic characteristics were analyzed using descriptive statistics. For continuous variables, such as age, the mean and standard deviation (SD) were estimated, and potential differences between the two groups were examined using a two-sample *t*-test or the non-parametric Mann–Whitney–Wilcoxon test, when appropriate. Analysis of variance (ANOVA) was used to compare more than two groups. Categorical variables were expressed as frequencies and proportions. Potential associations between two categorical variables (i.e., gender and HD group) were explored using a χ^2^ test of independence for contingency tables. If the frequency of a particular variable was low, a correction was made. Multivariate ANOVA (MANOVA) was utilized to assess the multiple dependency between two or more continuous variables of interest and potential predictors (i.e., HD diagnosis). In addition, correlations between two continuous variables were explored using Pearson’s correlation coefficient, *r*. In all cases, a *p*-value < 0.05 was considered to indicate statistical significance. Unless otherwise stated, statistical analyses and graphics were performed and generated using R version 4.0.3 [36].

## 5. Conclusions

In conclusion, our study included 291 participants, 33 of whom were diagnosed with Huntington’s disease (HD) based on the number of *HTT* CAG repeats. The most common genotype was 17/7_17/10. The genetic heterogeneity was quantified as the somatic mosaicism and slippage ratio, as implemented in ScaleHD. We found that the somatic mosaicism tends to increase with age in HD subjects, especially for the secondary allele, and that the slippage ratio for the secondary allele differed by the HD allele type. Our study provides insight into the genetic and molecular characteristics of HD in this population, which may inform future research and clinical management.

## Figures and Tables

**Figure 1 ijms-24-16154-f001:**
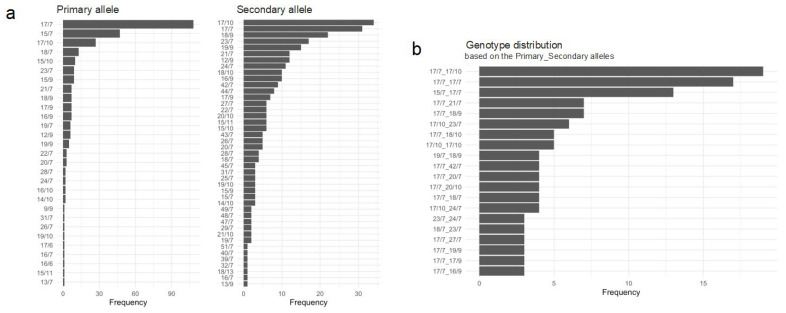
(**a**) Frequency distribution of the primary and secondary alleles; and (**b**) top 20 genotypes of the *HTT* gene in our cohort of 291 individuals from Juan de Acosta, Atlántico. In *a*/*b, a* is the number of *HTT* CAG repeats in the primary allele, and *b* is the number of *HTT* CCG repeats in the secondary allele.

**Figure 2 ijms-24-16154-f002:**
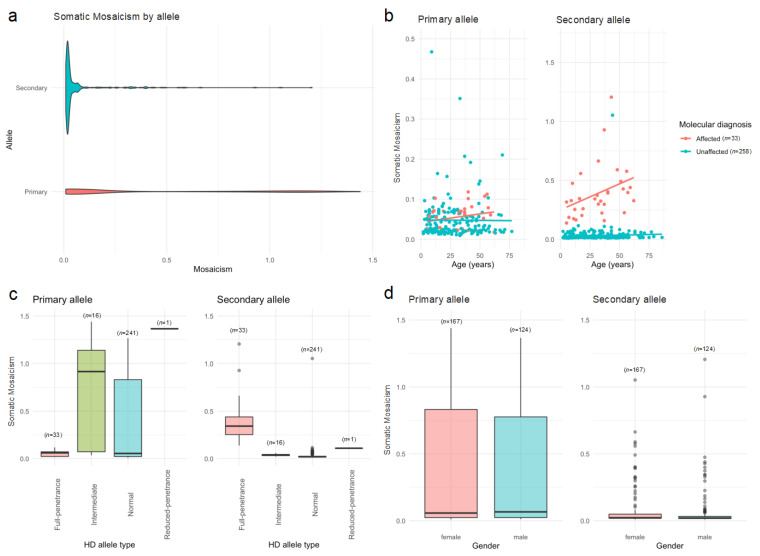
Heterogeneity as measured by the somatic mosaicism reported in ScaleHD, according to (**a**) allele, (**b**) age and molecular diagnosis, (**c**) primary and secondary alleles as a function of HD allele type, and (**d**) allele and sex in individuals from Juan de Acosta, Atlántico. Grey circles represent individuals’ observations.

**Figure 3 ijms-24-16154-f003:**
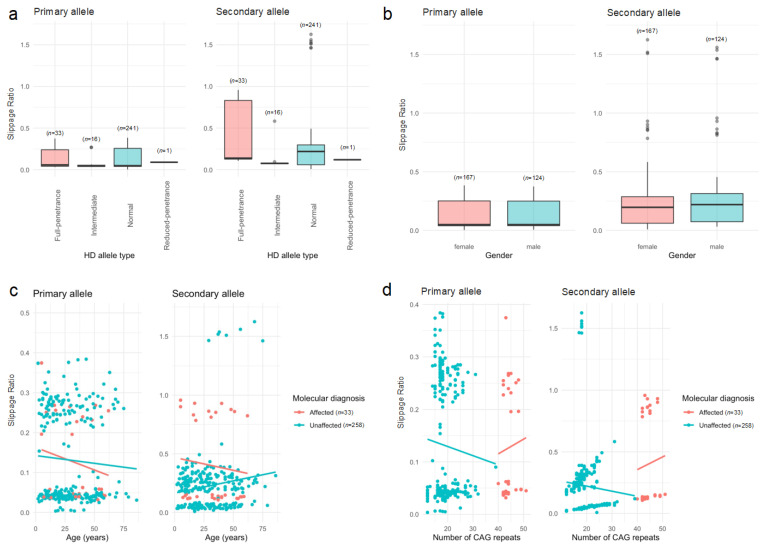
Heterogeneity as measured by the slippage ratio reported in ScaleHD, according to (**a**) allele; (**b**) allele and gender; (**c**) age and molecular diagnosis; and (**d**) number of CAG repeats in individuals from Juan de Acosta, Atlántico. Grey circles represent individuals’ observations.

**Figure 4 ijms-24-16154-f004:**
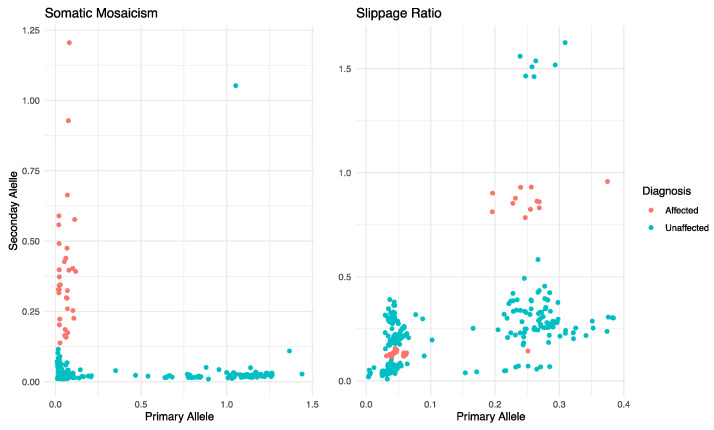
Scatterplots of the heterogeneity in the Primary and Secondary alleles as measured by the “Somatic Mosaicism” (**left**) and slippage ratio (**right**) according to molecular diagnosis.

**Table 1 ijms-24-16154-t001:** Sociodemographic characterization of individuals included in this study.

Variable	Affected *n* = 33 (11.3%)	Unaffected *n* = 258 (88.7%)	Statistic *^a^*	*p*-Value
Gender	Frequency (%)		0.03 (1)	0.867
Female	18 (10.8)	149 (89.2)		
Male	15 (12.1)	109 (87.9)		
Age	Frequency (%)		5.34 (3)	0.148
Children (<15 y)	8 (11.4)	62 (88.6)		
AYA (15–29 y)	4 (5)	76 (95)		
Adults (30–59 y)	19 (15.6)	103 (84.4)		
Elderly (>59 y)	2 (12.5)	14 (87.5)		
Schooling (years)	Frequency (%)		1 (4)	0.909
0	0 (0)	5 (100)		
1	2 (9.5)	19 (90.5)		
2	8 (11.8)	60 (88.2)		
3	13 (12.6)	90 (87.4)		
4	8 (13.8)	50 (86.2)		
HD type	Frequency (%)		525.87 (3)	<0.00001
Normal	-	241 (82.8)		
Intermediate	-	16 (5.5)		
Reduced penetrance	-	1 (0.3)		
Full penetrance	33 (11.3)	-		

*^a^* Results for the χ^2^ statistic of independence are shown. In parenthesis, the degrees of freedom (*df*) are reported. AYA: adolescents and young adults; HD: Huntington’s disease.

**Table 2 ijms-24-16154-t002:** Comparison of the number of *HTT* CAG repeats for normal and intermediate allele chromosomes between Colombian and other populations.

Population	Number of CAG Repeats					Reference
	Mean	SD	Range	*n*	*p*-Value	
Thai	16.5	1.9	8–28	449	Not reported	[10]
European	18.4	3.7	8–35	479	<0.0001	[12]
American	19.7	3.2	11–34	545	<0.0001	[13]
Finnish	17.1	1.8	14–23	48	0.255	[12]
Black	16.2	2.5	8–24	113	0.55	[12]
Chinese	16.4	1.5	8–20	90	1	[12]
Japanese	16.6	1.3	13–23	166	1	[12]
Colombian	18.2	3	12–35	257	0.00001	This study

SD: Standard deviation; *n*: sample size.

## Data Availability

The data presented in this study are available on reasonable request from the corresponding authors. The data are not publicly available due to the ongoing nature of the study and our commitment to protecting the privacy and confidentiality of our patients.

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
