# Peer review of "Uncovering the Genetic and Molecular Features of Huntington’s Disease in Northern Colombia"

_ijms, 2023, doi:10.3390/ijms242216154_

Round 1
Reviewer 1 Report
Comments and Suggestions for Authors
There are two things that are needed to be addressed.
1. Methods need to be clear and include more information about analyses.
2. Need to define terms. Two terms that are prominently used in this paper are used in different ways. Mosaicism is usually used to describe different genetic makeup in different tissues. If genetic variation arises from the same tissue type, heterogeneity will be more suitable. Also, the term primary and secondary alleles needed to be defined in the paper. This goes back to point 1. Since the authors did not specify how they analyzed data (i.e. whether to use CAG_1_1_CCG_2 or CAG_2_1_CCG_3), they might have got "secondary" alignment with different alignment parameters. Also, the slippage analysis is a very different assessment than what it is typically referred to as. Please specify how the slippage analysis has been conducted.

Not major, but need a good editor to proof-read.
Reviewer 2 Report
Comments and Suggestions for Authors
In this paper, Ahmad et al aimed to study the genetic and molecular characteristics of Juan de Acosta city inhabitants which are represented as the second most extensive pedigree clustering Huntington's disease (HD). Overall, I found the study interesting and have the potential to inform clinical management in this population and beyond.
Comments and suggestions:
1) Figures and tables should have a descriptive legend where N, significance test, etc are mentioned for readers to understand the data clearly.
2) Data visualization can be improved as it's hard to conclude results with some of the current figs. For example: in Fig 1 plot, it's very hard to interpret from the fig what the authors want to conclude. Probably more points on the x-axis where age distribution is plotted and clear labeling on y- the axis would make it more clear. or there is a lot of chart junk in the figs, better to revise the figs to make the results stand out.
3) Additionally, the paper could benefit from a more thorough discussion of the implications of the findings. This might include exploring the potential clinical applications of the research, highlighting areas for further study, or discussing how the results contribute to our understanding of HD and its genetic and molecular characteristics.
Round 2
Reviewer 1 Report
Comments and Suggestions for Authors
Thank you for addressing my concerns.
However, it is unclear whether the author sufficiently addressed the problems.
Please refer to them as "allele," not Genotype, as the authors of ScaleHD did in their own protocol, "We sort objects by a total number of aligned reads – the top allele is always taken as a ‘primary’ allele (quotation marks to note that ‘primary’ here does not have any biological meaning; it is simply the allele we primarily chose from the assembly)." The secondary allele, by design, refers to the second allele. I understand why authors refer them as "genotype" because ScaleHD output "Primary/Secondary GTYPE," but it is not genotype since genotype refers to both alleles. Also, it is unclear whether 17/7 means 17 CAG repeats with 7 CCG repeats or 17 CAG repeats allele with 7 CAG repeats allele. Based on Haplotype distribution, I can refer to them as 17 CAG repeats with 7 CCG repeats, but the authors never failed to explain this in the methods section. Also, since ScaleHD assumes the sequence will be "typical" structure of CAG_1_1_CCG_CCT, it would be nice to know whether there were any atypical CAG_2_1_7_3 structures were in the data.
Both mosaicism and slippage analysis will be invalid. Line 201 states "Slippage analyses in our sample revealed that slippage differed by HD allele type for the secondary allele, but not for the primary allele," which does not make biological sense. As ScaleHD output shows - BSlippage:: Slippage ratio of allele's read peak ('N minus 2' to 'N minus 1)', over 'N', and Somatic Mosaicism:: Mosaicism ratio of allele's read peak ('N plus 1' to 'N plus 10'), over 'N.' It has to be addressed as heterogeneity. The mere difference in repeat numbers does not show slippage or mosaicism.
Again, mosaicism refers to line 177-178 "genetic changes happening during early embryonic development, leading to some cells having one genetic variation while others have a different variation." The authors only collected blood samples, and within the blood sample, only leukocytes contain DNA, and all blood cells are differentiated from hematopoietic stem cells. The heterogeneity in DNA alignments can be raised from but not limited to - 1. fragmented DNA, 2. misalignment of the repeat sequence, 3. insufficient amplification and amplification preference to shorter allele. Also, ScaleHD output gives multiple alignments by default, so the authors of the paper should have considered filtering out the alignments.
Please report the actual configuration file as it reports quality score, minimum length of alignments, the maximum length of the alignment, error tolerance, sequence match score, mismatch penalty, indel penalty, gap extend penalty, primer clipping penalty, unpaired pairing, and sequence read quality cutoff.
There is a group missing from Figure 1 - no reduced-penetrance population distribution.
Comments on the Quality of English LanguageIt reads well but needs some editing and proofreading.
Author Response
Thank you for the opportunity to submit a revised version of our manuscript, " Uncovering the Genetic and Molecular Features of Huntington’s Disease in Northern Colombia”, which now includes responses to the concerns, enquiries, comments, and suggestions raised by two anonymous reviewers. Please find attached our response (in blue).

Round 3
Reviewer 1 Report
Comments and Suggestions for Authors
I thank the authors for addressing the comments.
Now the authors understand ScaleHD, the authors must agree that there is no reason to analyze primary alleles and secondary alleles separately. The high heterogeneity, "mosaicism," observed in the secondary allele is a mere artifact of sequencing alignment, not something that is biological. The reason why primary and secondary alleles exist in the first place is because the secondary allele did not align as well as the primary allele. The data will be much more insightful if the authors combine both primary and secondary alleles and analyze them together.
Somatic mosaicism and slippage analysis need to be combined into "heterogeneity analysis." ScaleHD referees one as an addition of repeats and the other one as subtraction. They do not have any biological meaning. Somatic mosaicism and slippage have very specific biological meanings, with more molecular analyses are needed to prove them. Authors can separate increased and decreased. It could be discussed in the discussion addition might have come from slippage and the decrease might have happened by repeat expansion during the mitosis process.
Comments on the Quality of English LanguageOkay.
Author Response
Dear reviewer,
Thank you so much for your insightful comments. We have revised it as you requested. Please see the attachment.
Best regards,
Authors
